# Comparative performance evaluation of Wako β-glucan test and Fungitell assay for the diagnosis of invasive fungal diseases

Elena De Carolis[1], Federica Marchionni[1], Riccardo Torelli[1], Morandotti Grazia Angela[1], Livio Pagano[2], Rita Murri[1], Gennaro De Pascale[3], Giulia De Angelis[1,4], Maurizio Sanguinetti[1,4]*, Brunella Posteraro[4,5]

**1** Dipartimento di Scienze di Laboratorio e Infettivologiche, Fondazione Policlinico Universitario A. Gemelli IRCCS, Rome, Italy, **2** Dipartimento di Diagnostica per Immagini, Radioterapia, Oncologia ed Ematologia, Fondazione Policlinico Universitario A. Gemelli IRCCS, Rome, Italy, **3** Dipartimento di Scienze dell'Emergenza, Anestesiologiche e della Rianimazione, Fondazione Policlinico Universitario A. Gemelli IRCCS, Rome, Italy, **4** Dipartimento di Scienze Biotecnologiche di Base, Cliniche Intensivologiche e Perioperatorie, Università Cattolica del Sacro Cuore, Rome, Italy, **5** Dipartimento di Scienze Gastroenterologiche, Endocrino-Metaboliche e Nefro-Urologiche, Fondazione Policlinico Universitario A. Gemelli IRCCS, Rome, Italy

* maurizio.sanguinetti@unicatt.it

**Data Availability Statement:** All relevant data are within the manuscript and its Supporting Information files.

## Abstract

The Fungitell assay (FA) and the Wako β-glucan test (GT) are employed to measure the serum/plasma 1,3-β-D-glucan (BDG), a well-known invasive fungal disease biomarker. Data to convincingly and/or sufficiently support the GT as a valuable alternative to the FA are yet limited. In this study, we evaluated the FA and the GT to diagnose invasive aspergillosis (IA), invasive candidiasis (IC), and *Pneumocystis jirovecii* pneumonia (PJP). The FA and GT performances were compared in sera of patients with IA (n = 40), IC (n = 78), and PJP (n = 17) with respect to sera of control patients (n = 187). Using the manufacturer's cutoff values of 80 pg/mL and 11 pg/mL, the sensitivity and specificity for IA diagnosis were 92.5% and 99.5% for the FA and 60.0% and 99.5% for the GT, respectively; for IC diagnosis were 100.0% and 97.3% for the FA and 91.0% and 99.5% for the GT, respectively; for PJP diagnosis were 100.0% and 97.3% for the FA and 88.2% and 99.5% for the GT, respectively. When an optimized cutoff value of 7.0 pg/mL for the GT was used, the sensitivity and specificity were 80.0% and 97.3% for IA diagnosis, 98.7% and 97.3% for IC diagnosis, and 94.1% and 97.3% for PJP diagnosis, respectively. At the 7.0-pg/mL GT cutoff, the agreement between the assays remained and/or became excellent for IA (95.1%), IC (97.3%), and PJP (96.5%), respectively. In conclusion, we show that the GT performed as well as the FA only with a lowered cutoff value for positivity. Further studies are expected to establish the equivalence of the two BDG assays.

## Introduction

Invasive aspergillosis (IA), invasive candidiasis (IC), and *Pneumocystis* pneumonia (PJP, previously known as PCP) represent the most prevalent invasive fungal diseases (IFDs) worldwide

**Funding:** M.S. received funding from FUJIFILM Wako Pure Chemical Corporation to perform this study. The funders had no role in study design, data collection and analysis, decision to publish, or preparation of the manuscript.

**Competing interests:** I declare that FUJIFILM Wako Pure Chemical Corporation provided reagents and funding for this study. This does not alter our adherence to PLOS ONE policies on sharing data and materials. FUJIFILM Wako Pure Chemical Corporation had no role in study design, data collection and analysis, decision to publish, or preparation of the manuscript.

[1]. These diseases mainly affect immunocompromised (or immunosuppressed) hosts, causing estimated over 1.6 million deaths annually [2]. Causative agents of IC include different *Candida* species [3], whereas the main cause of IA remains *Aspergillus fumigatus* [4] and PJP is uniquely caused by *Pneumocystis jirovecii*, formerly *Pneumocystis carinii* [5]. As IFD symptoms can be subtle and/or nonspecific, it is difficult to identify and treat the cause of disease, especially in patients with hematological malignancies [6]. Furthermore, microbiological confirmation of IFD with conventional, culture-dependent methods may yield false-negative results [7], hence, molecular, culture-independent methods to enhance the diagnostic sensitivity need to be developed [8]. Thus, pending the microbiological diagnosis, an empirical treatment targeting infectious and non-infectious causes may be necessary [6].

As broad fungal biomarker for IFD (the only notable exception are mucormycosis, cryptococcosis, and blastomycosis) [9–12], serum 1,3-β-D-glucan (BDG) has shown wide utility in specific clinical settings [13], including IA, IC, and PJP [14]. In a meta-analysis, He *et al.* focused on the accuracy of cutoff values to diagnose IFD obtained with BDG detection assays [15]. One was the Fungitell assay (FA; Associates of Cape Cod, East Falmouth, MA), FDA cleared and Conformité Européenne (CE) marked, which has been most used in the Western Hemisphere (Europe and United States), and another was the Wako β-glucan test (GT; FUJIFILM Wako Pure Chemical Corporation, Osaka, Japan), CE marked, which has recently been introduced in Europe.

Both assays rely on the BDG ability to activate factor G, a serine protease zymogen, in the *Limulus* (horseshoe crab) coagulation cascade. Activated factor G converts the inactive proclotting enzyme to the active form, which in turn cleaves an artificial substrate used for colorimetric (FA) or turbidimetric (GT) detection. Although BDG concentrations are measured through spectrophotometric readings, dissimilarity of cutoff values between the assays may be related to differences in the standards and/or affinity/reactivity of reagents in each assay [16]. Using the proposed 80 pg/mL (FA) and 11 pg/mL (GT) cutoff values [15], a recent comparison of the two assays for PJP diagnosis showed GT to be more specific and FA to be more sensitive, at a statistically significant level [17]. Interestingly, the sensitivity of GT equaled that of FA (at a cutoff of $\geq$60 pg/mL) and the specificity was significantly better than that of the FA, when the GT cutoff value lowered from 11 pg/mL to 3.616 pg/mL [17]. Consistently, previous work showed sensitivities of the FA for both IC (i.e., candidemia) and PJP diagnoses to be superior to those of the GT [18]. Again, lowering the GT cutoff value to $\geq$3.8 pg/mL resulted in sensitivities of the GT that became acceptable for candidemia (with a decreasing specificity from 98.0% to 91.0%) and excellent for PJP, respectively [18]. However, while more data are required to support the GT as a valuable alternative to the FA (especially for patients with candidemia), not enough data exist about the GT to diagnose IA.

Therefore, we compared the performance of the GT with that of the FA in well-characterized groups of patients with IA, IC, and PJP, with reference to appropriate control patients. Similar to previous studies [17, 18], we also tried to define the optimal GT cutoff values which could allow to reliably exclude IFD (mainly due to *Aspergillus*, *Candida*, or *P. jirovecii*).

## Materials & methods

### Ethics statement

This study was conducted at the Fondazione Policlinico Universitario A. Gemelli (FPG) IRCCS of Rome, Italy, and was approved by the Ethics Committee of the FPG (application number 38367/19) and a waiver of informed consent was granted. Sample processing and data analysis were performed anonymously.

## Study design

We conducted a retrospective performance assessment study on archived patients' serum samples, which had been collected as part of routine clinical care at the FPG IRCCS, a large tertiary care hospital in Rome, Italy. We used serum instead of plasma samples because of our previous findings showing that the two sample types may be considered equivalent [19]. Samples were from adult patients (mainly oncology/hematology patients) who were classified as having proven or probable IFD according to the 2008 European Organization for the Research and Treatment of Cancer/Mycoses Study Group (EORTC/MSG) criteria [20]. Samples were also from patients who were classified as having PJP if they had clinical signs and symptoms of respiratory infection (i.e., dyspnea, cough, or hypoxemia) supported with radiological findings (e.g., diffuse ground glass opacities on chest radiograph) and with positive *Pneumocystis*-specific PCR and/or immunofluorescence staining results of respiratory samples [21]. To enhance the robustness of the analysis, only patients with proven or probable IFDs were evaluated as IFD cases. Serum samples from patients who had major risk factors for IFD but who did not meet the criteria for proven or probable disease (i.e., who had no evidence of IFD) were included as controls. Furthermore, no diagnoses were based on detection of BDG (see below), which is part of the mycological criteria for the 2008 EORTC/MSG classification of patients with proven/probable/possible or no IFD [20]. At the time of writing this article—and shortly after completion the study, the EORTC/MSG education and research consortium updated definitions of IFDs [22]. However, according to the 2019 EORTC/MSG criteria, no reclassification was required for either IA or IC, while PJP cases were all classifiable as proven IFD, in our study [22].

We initially considered all the patients with a first serum BDG sample collected for routine mycological testing eligible. Then, we included only the patients for whom a classification as proven or probable IA (or IC) according to the 2008 EORTC/MSG definitions was available. All patients' samples were from the closest time (± 2 days) when evidence or no evidence of IFD was obtained for cases and controls, respectively, and were stored at –80°C until the study time.

## Serum BDG measurement

Before BDG testing, frozen serum samples were thawed at room temperature and briefly vortexed. The same investigator tested each sample's aliquots with the FA and the GT in parallel and in a blinded manner for each patient's disease classification at the time of testing. The two assays were performed in accordance with the manufacturer's instructions. In both assays, the BDG measurement relies on a modification of the *Limulus* amebocyte lysate (LAL), which results in colorimetric (FA) or turbidimetric (GT) reaction changes. Briefly, for the FA 5 μL of serum (in duplicate) was used and the LAL reaction was monitored at 37°C for 40 min in an ELx808 microplate reader (BioTek Instruments, Winooski, VT). By comparing with a standard curve, the mean optical density change over time was calculated to determine the sample's BDG concentration. A positivity threshold of 80 pg/mL was used throughout the study. For the GT, 100 μL of serum was used and the LAL reaction (i.e., gelation) was monitored at 37°C for a maximum of 90 min in a MT-6500 toxinometer (FUJIFILM Wako Pure Chemical Corporation). By comparing with a calibration curve (supplied with each lot by the manufacturer), the gelation time was calculated to determine the sample's BDG concentration. A positivity threshold of 11 pg/mL was initially used in the study (see below for positivity threshold optimization). Samples with positive results of > 500 pg/mL (FA) or > 600 pg/mL (GT) were diluted and retested.

## Data collection and statistical analysis

Statistical analysis was performed using GraphPad Prism version 8.2 (GraphPad Software, La Jolla, CA) and MedCalc Statistical Software version 19.0.7 (MedCalc Software bvba, Belgium). Patients' demographics and BDG data were expressed as percentages or medians with inter-quartile ranges (IQR), respectively. To determine sensitivity and specificity of BDG assays, together with their respective 95% confidence intervals (CIs), we constructed 2 × 2 tables using IFD (IA, IC, or PJP) patients as true cases and non-IFD patients as controls. For both assays, receiver operating characteristic (ROC) curves were generated and cutoff values were derived to assess the diagnostic accuracy of serum BDG for case (IFD) versus control (non-IFD) patients. The highest Youden index indicated the optimal BDG cutoff. Variables were compared using chi-square test and Mann-Whitney U test, as appropriate, whereas differences in performance parameters between BDG assays were assessed using McNemar's test. Agreement between BDG assays was determined by Spearman's correlation, whereas the strength of agreement was determined by Cohen's kappa statistic. Thus, values higher than 0.80 represented excellent agreement, values between 0.80 and 0.4 represented substantial to moderate agreement, and values lesser than 0.4 represented fair to slight agreement. Statistical significance was set at a $< .05$ $P$-value.

## Results

### Study samples

Table 1 shows demographics for the 322 patients from whom 322 serum samples were tested with two BDG assays (FA and GT), of which 78 samples were from proven IC (75 candidemia and 3 intraabdomonary candidiasis) cases, 40 samples from probable IA (38 pulmonary and 2 disseminated aspergillosis) cases, and 17 samples from PJP cases. For all 135 cases, fungal disease classification (proven or probable) was obtained as previously described [20, 21]. We also included 187 samples from 187 patients, for whom clinical, radiographic, and microbiological findings indicated no evidence of IFD (controls). Table 2 shows details about the mycological evidence used to classify the 135 cases as IFD.

### Performances of the FA and the GT

The FA (cutoff for positivity, ≥80 pg/mL) and the GT (cutoff for positivity, ≥11 pg/mL) gave positive results in 132 (97.8%) and 110 (81.5%) of 135 IFD samples, respectively. Of 187

**Table 1. Characteristics of 322 patients from whom serum samples were tested for BDG measurement[a].**

|  | Value for patients with: | | | |
|---|---|---|---|---|
|  | IA (n = 40) | IC (n = 78) | PJP (n = 17) | No IFD (n = 187) |
| Median (interquartile range) age, years | 53.0 (38.7–60.2) | 52.5 (40.2–60.0) | 52.0 (33.0–64.0) | 54.0 (42.0–68.0) |
| Sex, male/female | 18/22 | 34/44 | 7/10 | 90/97 |
| Underlying condition, no. of patients (%) |  |  |  |  |
| Abdominal surgery | 0 (0.0) | 6 (7.7) | 0 (0.0) | 15 (8.0) |
| Solid tumor | 9 (22.5) | 22 (28.2) | 2 (11.8) | 23 (12.3) |
| Haematologic malignancy/HSCT | 23 (57.5) | 2 (2.6) | 7 (41.2) | 92 (49.2) |
| Other[b] | 8 (20.0) | 48 (61.5) | 8 (47.0) | 57 (30.5) |

BDG, 1,3-β-D-glucan; IA, invasive aspergillosis; IC, invasive candidiasis; PJP, *Pneumocystis jirovecii* pneumonia; IFD, invasive fungal disease; HSCT, hematopoietic stem cell transplantation.

[a] BDG testing was performed in parallel with the Fungitell assay (FA) and the Wako β-glucan test (GT).

[b] Includes patients in intensive care and infectious disease units.

**Table 2. Mycological characteristics of IFD cases[a].**

| Characteristic | IA cases (N = 40) | | IC cases (N = 78) | | PJP cases (N = 17) | |
|---|---|---|---|---|---|---|
| | n | % | n | % | n | % |
| Proven IFD | | | | | | |
| By culture | – | – | 75 | 96.1 | NA | – |
| By histology | – | – | 3 | 3.8 | NA | – |
| By immunofluorescence | NA | – | NA | – | 17[b] | 100 |
| Probable IFD | | | | | | |
| By galactomannan antigen tested in: | | | | | | |
| Serum | 37[c] | 92.5 | NA | – | NA | – |
| BALF | 5[d] | 12.5 | NA | – | NA | – |
| CSF | 2[e] | 5.0 | NA | – | NA | – |

NA, Not applicable; BALF, bronchoalveolar lavage fluid; CSF, cerebrospinal fluid.

[a] All IFD cases were classified as described in the text.

[b] 15 cases were also PJP-specific PCR positive.

[c] The median (interquartile range) value of galactomannan index (optical density [OD] of sample/OD of cutoff control) was 1.6 (1.3–2.1). Indices of $\geq 0.5$ were considered positive.

[d] Two cases had positive galactomannan detection results also in serum samples. Indices were 0.8 (serum) and 1.5 (BALF) in one case and 0.9 (serum) and 2.2 (BALF) in the other case.

[e] Two cases had positive galactomannan detection results also in CSF samples.

control samples, 182 (97.3%) and 186 (99.5%) had negative results with the FA and the GT, respectively. The one patient with a false-positive result by the GT was highly positive by the FA. Overall, the median BDG concentration was 398 pg/mL with the FA and 31.63 pg/mL with the GT. In IA cases, the median (IQR) BDG concentrations were 201 (119–377) pg/mL and 14.07 (7.45–41.46) pg/mL for the FA and the GT, respectively. These levels were significantly higher than the levels in controls (0 (0–11) and 0 (0–0), respectively; $P < .0001$). Using the aforementioned cutoff values, the percentages of sensitivity and specificity were 92.5 and 99.5 for the FA, and 60.0 and 99.5 for the GT, respectively (Table 3). The sensitivity of FA was significantly greater than the sensitivity of GT (92.5% versus 60.0%, $P < .001$).

In IC cases, the median (IQR) BDG concentrations were 516 (250–837) pg/mL and 45.57 (16.10–109.9) pg/mL for the FA and the GT, respectively. These levels were significantly higher than the levels in controls ($P < .0001$). Using the aforementioned cutoff values, the percentages of sensitivity and specificity were 100.0 and 97.3 for the FA, and 91.0 and 99.5 for the GT, respectively (Table 3). The sensitivity of FA was significantly greater than the sensitivity of GT (100.0% versus 91.0%, $P < .001$).

In PJP cases, the median (IQR) BDG concentrations were 512 (404–639) pg/mL and 46.88 (26.43–166.3) pg/mL for the FA and the GT, respectively. These levels were significantly higher than the levels in controls ($P < .0001$). Using the aforementioned cutoff values, the percentages of sensitivity and specificity were 100.0 and 97.3 for the FA, and 88.2 and 99.5 for the GT, respectively (Table 3). The sensitivity of FA was significantly greater than the sensitivity of GT (100.0% versus 88.2%, $P = .01$).

To improve the GT diagnostic performance, we determined the optimal positivity threshold for the GT (7.0 pg/mL) with the highest Youden index, which corresponds to the maximal sensitivity and specificity combination. As depicted in Fig 1, ROC analysis for proven/probable IFD versus no IFD at the optimized cutoff value generated an area under the curve (AUC) of 0.992 (95% CI, 0.975–0.999). Using the 7.0-pg/mL cutoff, the percentages of sensitivity and

**Table 3. Performance of the Fungitell assay and the Wako β-glucan test using indicated cutoffs to distinguish between IFD and non-IFD patients[a].**

| Parameter | Fungitell assay result (cutoff of ≥80 pg/mL) | Wako β-glucan test result (cutoff of ≥11 pg/mL) | Wako β-glucan test result (cutoff of ≥7 pg/mL) |
|---|---|---|---|
| Invasive candidiasis, n = 78 | | | |
| True positives | 78 | 71 | 77 |
| False negatives | 0 | 7 | 1 |
| True negatives | 182 | 186 | 182 |
| False positives | 5 | 1 | 5 |
| Sensitivity, % (95% CI[b]) | 100.0 (95.3–100.0) | 91.0 (82.6–95.5) | 98.7 (93.0–99.9) |
| Specificity, % (95% CI) | 97.3 (93.8–98.8) | 99.5 (97.0–99.9) | 97.3 (93.8–98.8) |
| Invasive aspergillosis, n = 40 | | | |
| True positives | 37 | 24 | 32 |
| False negatives | 3 | 16 | 8 |
| True negatives | 182 | 186 | 182 |
| False positives | 5 | 1 | 5 |
| Sensitivity, % (95% CI) | 92.5 (80.1–97.4) | 60.0 (44.6–73.6) | 80.0 (65.2–89.5) |
| Specificity, % (95% CI) | 97.3 (93.8–98.8) | 99.5 (97.0–99.9) | 97.3 (93.8–98.8) |
| *Pneumocystis jirovecii* pneumonia, n = 17 | | | |
| True positives | 17 | 15 | 16 |
| False negatives | 0 | 2 | 1 |
| True negatives | 182 | 186 | 182 |
| False positives | 5 | 1 | 5 |
| Sensitivity, % (95% CI) | 100.0 (81.5–100) | 88.2 (65.6–97.9) | 94.1 (73.0–99.7) |
| Specificity, % (95% CI) | 97.3 (93.8–98.8) | 99.5 (97.0–99.9) | 97.3 (93.8–98.8) |

[a] Serum samples from 135 patients with invasive fungal disease (IFD) and from 187 patients without evidence of IFD (non-IFD controls) were tested with the Fungitell assay (FA) and the Wako β-glucan test (GT). The manufacturers' cutoff values (FA, ≥80 pg/mL, and GT, ≥11 pg/mL) and, only for the GT, the optimized cutoff value (≥7 pg/mL) were used as sample's positivity thresholds.

[b] CI, confidence interval.

specificity for the diagnosis of proven/probable IFD were 80.0 and 97.3 for IA, 98.7 and 97.3 for IC, and 94.1 and 97.3 for PJP, respectively (Table 3). Only for IA, the sensitivity of FA remained significantly greater than the sensitivity of GT (92.5% versus 80.0%, $P < .03$). Similarly, we performed a ROC analysis for the FA using an optimized cutoff of 79.0 pg/mL, and this resulted in an AUC of 0.990 (95% CI, 0.972–0.998). Additionally, we tried to define whether a different cutoff value would be required for IA compared to IC or PJP. The values for GT determined with the highest Youden index were 2.6 pg/mL for IA and 7.0 pg/mL for both IC and PJP, whereas those for FA were 79 pg/mL for IA, 105 pg/mL for IC, and 88 pg/mL for PJP.

## Comparability of the FA and GT results

Assessing the quantitative agreement between the assays showed that a significant correlation between the BDG levels measured by the FA and the GT for IFD and non-IFD patient samples (Fig 2). In particular, these levels correlated significantly—although differently—for the IA ($r = 0.670$ and $P < .0001$; n = 40), IC ($r = 0.703$ and $P < .0001$; n = 78), and PJP ($r = 0.667$ and

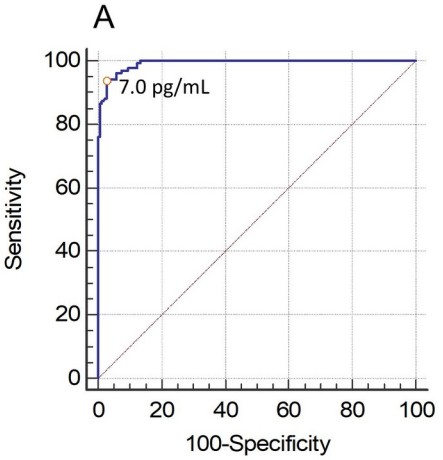
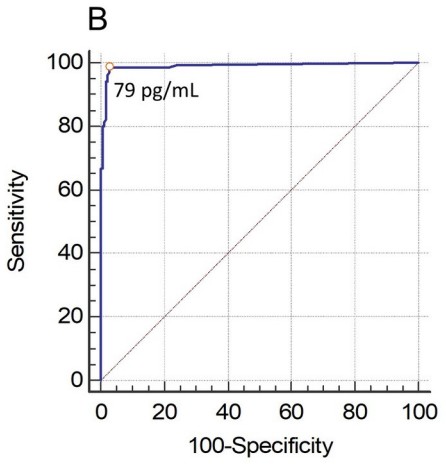

| | AUC (95% CI) | Optimal cutoff | *P*-value |
|---|---|---|---|
| GT | 0.992 (0.975–0.999) | ≥ 7.0 pg/ml | < 0.0001 |
| FA | 0.990 (0.972–0.998) | ≥ 79 pg/ml | < 0.0001 |

**Fig 1. Receiver operating characteristic (ROC) curves of the GT (A) and the FA (B).** The optimized thresholds for positivity in both assays (7.0 pg/mL and 79 pg/mL, respectively) are marked with a dot.

*P* = .004; n = 17) patient samples. Assessing the qualitative agreement between the assays showed that an excellent (or substantial) concordance regarding the results (positive/negative) obtained with the FA (cutoff for positivity, ≥80 pg/mL) and GT (cutoff for positivity, ≥11 pg/mL). This occurred with the samples from patients with IFD (91.8% agreement; Cohen's kappa statistic, 0.82), IA (91.6% agreement; Cohen's kappa statistic, 0.67), IC (95.0% agreement; Cohen's kappa statistic, 0.88), and PJP (96.0% agreement; Cohen's kappa statistic, 0.76), respectively. Using the optimized GT cutoff value (≥7.0 pg/mL), the concordance between the assays remained and/or became perfect for the samples from patients with IFD (95.6% agreement; Cohen's kappa statistic, 0.91), IA (95.1% agreement; Cohen's kappa statistic, 0.83), IC (97.3% agreement; Cohen's kappa statistic, 0.93), and PJP (96.5% agreement; Cohen's kappa statistic, 0.81), respectively.

## Discussion

To the best of our knowledge, two retrospective case-control studies have carefully evaluated the GT—the last approved BDG assay on the European market of s—in comparison with the FA in serum samples [17, 18]. Using a similar study design, we compared the GT with the FA to diagnose IFD in 322 patients categorized into IC (n = 78), IA (n = 40), PJP (n = 17), and non-IFD (n = 187) groups. Indeed, we established the overall GT and FA performances in two patient groups (135 with IFD and 187 without IFD), as well as the specific GT and FA performances in the IA, IC, and PJP groups with respect to the non-IFD group. Importantly, to compare the GT with the FA results, we followed the strategy of testing all archived patient samples in parallel, which reduces the bias due to probability of BDG degradation during sample storage, as successfully shown [17]. In contrast, Friedrich *et al.* performed FA measurement at the time of sampling and GT measurement on long-term stored samples [18].

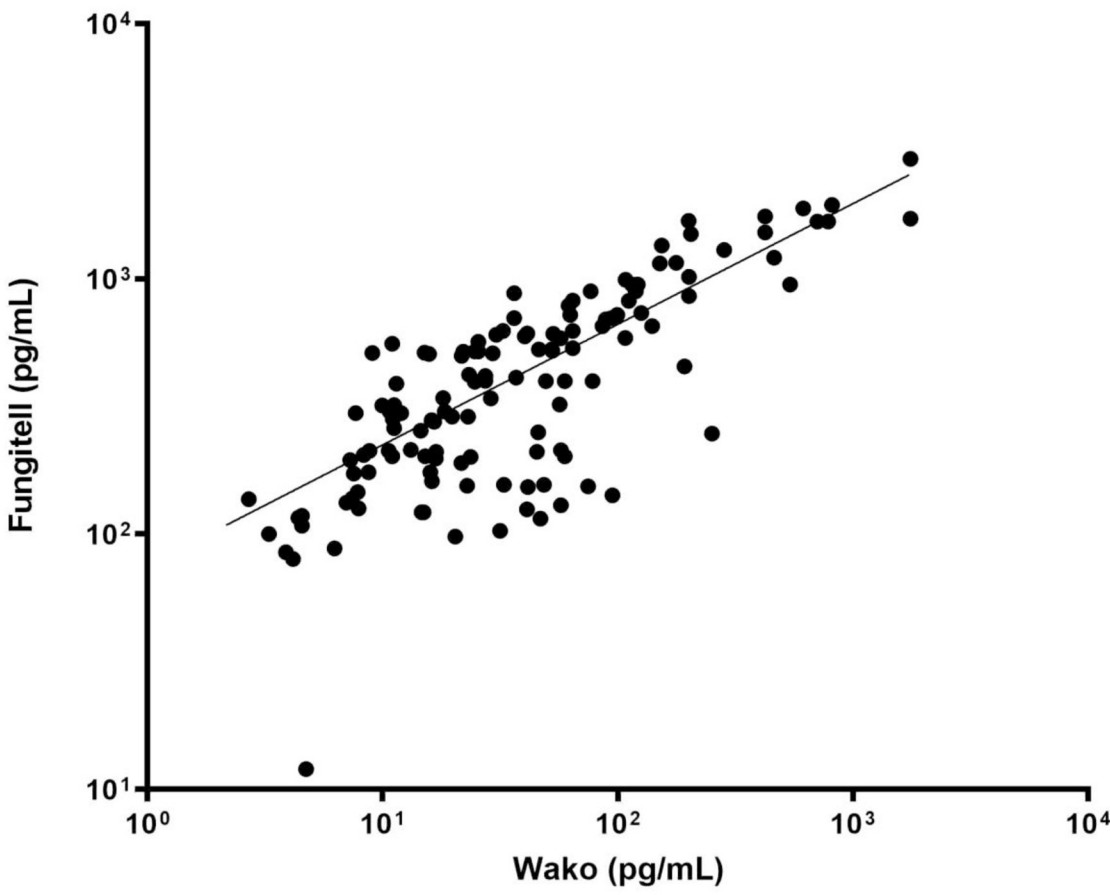

Spearman $r$ = 0.9327, p < 0.0001

**Fig 2. Correlation between the BDG concentrations (pg/mL) determined by the FA and GT assays.**

By applying the manufacturer's recommended cutoffs for positivity, we showed that the sensitivity of the FA was higher than the sensitivity of the GT in IFD patients overall (97.8% versus 81.5%, respectively) and, in particular, IA (92.5% versus 60.0%, respectively), IC (100% versus 91.0%, respectively), or PJP (100% versus 88.2%, respectively) patients. While the PJP values in our study are very similar to those reported by Friedrich *et al.* in PJP patients (sensitivity of FA and GT were 100% and 88.9%, respectively) [18], our IC values differ substantially from those reported by Friedrich *et al.* in candidemia patients (sensitivity of FA and GT were 86.7% and 42.5%, respectively) [18]. However, our findings are surprising, particularly if compared to the results of several meta-analyses published over recent years [9–11]. The pooled sensitivity and specificity of BDG for invasive fungal infections (excluding *P. jirovecii* infections) varied from 76.0% and 85.0% [10], 76.8% and 85.3% [9] to 80.0% and 82.0% [11], respectively. Conversely, in one meta-analysis [11], the pooled sensitivity and specificity for PJP were 96.0% and 84.0%, respectively.

In agreement with the results by meta-analyses [9–11] but in marked contrast with those by Friedrich *et al.* [18], one study [23] showed a sensitivity of 67.0%—neighbor, albeit still distant, to ours—and a specificity of 93.0% for the GT in a similar-sized cohort of patients with candidemia. Like in the Friedrich *et al.*'s study [18], applying a decreased cutoff of 7 pg/mL allowed Dichtl *et al.* to obtain a sensitivity of 73.0%, while specificities in both the studies remained as high as 91.0% [18] and 93.0% [23]. Another study by Dichtl *et al.* [24] conducted in the setting of PJP, showed that the specificity of GT—using an 11-pg/mL cutoff—was 100% in 25 control individuals who tested negative for *P. jirovecii* DNA by quantitative real-time PCR from respiratory samples. In the same study [24], the GT sensitivity increased from 86.0% to 91.0% after exclusion of the cases with slightly positive PCR results—that are usually negative in microscopy. Thus, our data unravel unexpected features of both GT and FA, which seem to subvert the paradigms of specificity as a "FA weakness" and of sensitivity as a "GA weakness" [25].

The reasons for conflicting results among studies are unknown, but we cannot exclude the influence of factors as a possible explanation for the observed differences in BDG assays' sensitivity and specificity. For example, findings from stratified (subgroup) analyses in two independent meta-analysis studies led to observe a lower EORTC/MSG diagnostic accuracy attributable to lower sensitivity compared to similar criteria (e.g., histopathological examination and/or microbiological culture from blood or sterile material) [11, 15]. In the present study, we used the 2008 EORTC/MSG criteria for both IA and IC [20], and 3 (1.3%) of 78 patients had an IC diagnosis based on histology only. Unlike us, Friedrich *et al.* [18] exclusively used mycological culture from blood as a reference standard. Although blood culture serves to document the proven presence of invasive fungal infection [20], three patients with false-positive BDG results in the study by Friedrich *et al.* had negative blood cultures but were positive for the mannan antigen [18], which is a highly specific serum biomarker for IC [26]. Furthermore, fungal antigen levels in the blood are generally higher in culture-positive cases, so the inclusion of patients based on a blood culture positive may not allow a correct appraisal of the BDG performance [23]. In our study, the median (IQR) value of galactomannan OD index in 40 patients diagnosed with probable IA was 1.6 (1.3–2.1), giving a measure of moderately high fungal burdens in these patients. This could indicate later infection stages in these patients— that in turn could result in higher BDG values—unless the exposure to mold-active antifungal prophylaxis in many patients could have masked their actual galactomannan levels [8]. Ultimately, for all analyses, we used a mixed control group of patients with different (albeit partially overlapping) risk factors for IFD [8, 27], which could skew the BDG diagnostic parameters. To exclude this possibility, we performed a rough analysis using control subgroups. Interestingly, we found no one or slight differences in specificities when they were calculated with patients only at risk of IC (97.6% [FA] and 99.4% [GT]), IA (100% for both FA and GT), or PJP (99.1% [FA] and 100% [GT]) (data not shown).

Unlike IA, where the low sensitivity (60.0%) makes the GT not suitable for diagnosis using the manufacturer's recommended cutoff value of 11 pg/mL, the sensitivities of the GT in IC (91.0%) and PJP (88.2%) were already good without lowering the cutoff value. However, the GT sensitivity in IA increased to 80.0% with a positivity threshold of 7.0 pg/mL and without compromising specificity (97.3%). Applying the 7.0-pg/mL cutoff value, the GT sensitivity also increased for both IC (from 91.0% to 98.7%) and PJP (from 88.2% to 94.1%), whereas at this cutoff the specificity for both IC and PJP slightly decreased (from 99.5% to 97.3%). Interestingly, the GT cutoff value proposed in this study was almost twice the value proposed elsewhere, such as 3.8 in candidemia [18] and 3.616 in PJP [17]. Of note, our value also encompasses IA, and this is not surprising because BDG testing for yeast or yeast-like fungi such as *Candida* species and *P. jirovecii* may require a cutoff value different from that required for molds such as *Aspergillus* species. Accordingly, cutoffs to optimize the performance of the

BDG assay may be different depending on the pathogen and host [14]. Therefore, it should be mandatory to interpret BDG results in relation to not only the specific pathogen, but also the specific patient population [28].

Our comparison of two BDG assays showed that both quantitative correlation and qualitative agreement between the FA and the GT were very good. However, one of the assays failed to categorize patients correctly. In particular, 13 IA, 7 IC, and 2 PJP samples provided false negative results only with the GT, because the BDG content in these samples did not reach the positivity threshold recommended by the manufacturer. Lowering the GT cutoff value allowed us to recover 5 IA, 6 IC, and 1 PJP samples as true positive samples, thereby indicating that the manufacturer's cutoff value might not be appropriate for patients with IA, IC, and PJP. This was in agreement with the observations reported elsewhere [17, 18].

With the present study, we aimed to provide further data about the role of serum BDG in the diagnosis of IFDs by means of a comparative assessment of both assays in terms of sensitivity and specificity using a large number of patient samples. However, we acknowledge the unavoidable limitations of the study. First, we did not explore whether lower sensitivity of the GT, particularly in IA samples, could be due to the use of serum instead of plasma, which the GT manual has listed as a principal specimen for clinical investigation. Nevertheless, we previously showed that serum samples effectively equate plasma samples for BDG measured with GT in patients with probable or proven fungal diseases [19]. Second, we did not assess the predictive value of both assays because of the artificially high prevalence of IFD in our test population influencing the value. However, we well know that, at least for candidemia, high sensitivity and negative predictive value for serum BDG are both helpful to exclude disease in order to withhold treatment [29]. Third, we waived to compare both assays for important features such as their layout and workflow. However, we experimented that GT is technically less complex than the FA regarding the possibility either of testing samples individually or up to 16 samples in parallel, as well as the use of a standard curve provided by the manufacturer and, above all, more simplicity to execute the assay. Fourth, unlike previous studies [17, 18], the number of PJP case tested by us was very small.

In conclusion, while confirming the good diagnostic accuracy of serum BDG assay [30], our findings support and/or extend the diagnostic value of both FA and GT into clinical settings such as IA, IC, and PJP. We show that the GT performed almost as the FA after optimizing the GT cutoff value for positivity. Further studies are yet necessary to establish the equivalence of both assays in order to provide equally reliable BDG results that can ultimately help clinicians with hard-to-diagnose fungal diseases.

## Supporting information

**S1 Data.**
(CSV)

**S2 Data.**
(CSV)

**S3 Data.**
(CSV)

**S4 Data.**
(CSV)

## Author Contributions

**Conceptualization:** Elena De Carolis, Maurizio Sanguinetti, Brunella Posteraro.

**Data curation:** Riccardo Torelli, Morandotti Grazia Angela, Rita Murri, Gennaro De Pascale, Giulia De Angelis, Maurizio Sanguinetti.

**Formal analysis:** Elena De Carolis, Federica Marchionni, Riccardo Torelli, Livio Pagano, Rita Murri, Gennaro De Pascale, Giulia De Angelis.

**Funding acquisition:** Maurizio Sanguinetti.

**Methodology:** Morandotti Grazia Angela, Livio Pagano, Giulia De Angelis, Brunella Posteraro.

**Supervision:** Maurizio Sanguinetti, Brunella Posteraro.

**Validation:** Elena De Carolis, Brunella Posteraro.

**Writing – original draft:** Brunella Posteraro.

**Writing – review & editing:** Maurizio Sanguinetti.

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
