## [Decision Letter · Decision Letter 0]

1 Apr 2020

PONE-D-19-33241

Comparative performance evaluation of Wako β-glucan test and Fungitell assay for the diagnosis of invasive fungal diseases including Pneumocystis pneumonia

PLOS ONE

Dear Dr. Sanguinetti,

Thank you for submitting your manuscript to PLOS ONE. After careful consideration, we feel that it has merit but does not fully meet PLOS ONE’s publication criteria as it currently stands. Therefore, we invite you to submit a revised version of the manuscript that addresses the points raised during the review process.

I would ask that you address the comments raised by the reviewers, in particular I would ask that you fully address the comments and questions raised regarding the significant improvements of the Glucan Test and Fungitell Assay compared to the published literature.

We would appreciate receiving your revised manuscript by May 14 2020 11:59PM. To enhance the reproducibility of your results, we recommend that if applicable you deposit your laboratory protocols in protocols.io, where a protocol can be assigned its own identifier (DOI) such that it can be cited independently in the future. For instructions see: http://journals.plos.org/plosone/s/submission-guidelines#loc-laboratory-protocols

We look forward to receiving your revised manuscript.

Kind regards,

Jeffrey Chalmers, Ph.D.

Academic Editor

PLOS ONE

"M.S. received funding from FUJIFILM Wako Pure Chemical Corporation to perform this study. The funders had no role in study design, data collection and analysis, decision to publish, or preparation of the manuscript."

We note that you received funding from a commercial source: FUJIFILM Wako Pure Chemical Corporation which produces the Wako Beta-Glucan test.

Reviewers' comments:

Reviewer's Responses to Questions

**Comments to the Author**

1. Is the manuscript technically sound, and do the data support the conclusions?

Reviewer #1: Yes

Reviewer #2: Yes

Reviewer #3: Partly

Reviewer #4: Yes

2. Has the statistical analysis been performed appropriately and rigorously? 

Reviewer #1: I Don't Know

Reviewer #2: Yes

Reviewer #3: N/A

Reviewer #4: Yes

3. Have the authors made all data underlying the findings in their manuscript fully available?

Reviewer #1: Yes

Reviewer #2: Yes

Reviewer #3: Yes

Reviewer #4: No

4. Is the manuscript presented in an intelligible fashion and written in standard English?

Reviewer #1: Yes

Reviewer #2: No

Reviewer #3: Yes

Reviewer #4: No

5. Review Comments to the Author

Reviewer #1: This article addresses the diagnostic cutoffs for the available fungal diagnostic tests. It suggests new cutoffs that improve detection for those tests, which is in agreement with previous data. The change in cutoff is very small however. This article may be relevant for the field.

Reviewer #2: Thank you for the opportunity to review this manuscript.

Carolis and colleagues present a well- performed study demonstrating the performance of the Wako beta glucan assay in comparison to the Fungitell assay. Overall, the study design is robust and performance of the study is well explained.

English language editing can be sought to improve the readability of the manuscript.

Specific comments:

- Line 45: suggest reword the opening sentence to: "Invasive Aspergillosis, IC and PJP represent the most prevalent invasive fungal infections worldwide".

- Line 46: reword "previously known as" to "also known as". PCP is still being used as a term for the disease

- Line 51: Suggest remove "e.g. respiratory"

- Line 51: non-specific, not unspecific

- Line 54: Suggest reword- instead of unsuccessful, rather mention that culture methods have poor sensitivity or yield false negative results

- Line 56: reword as empiric treatment may be necessary. (the reference used here relates specifically to respiratory disease in patients with haematological disease). In many other cases of invasive fungal disease, it may not be appropriate to initiate empiric therapy

- Line 57: Reword as a "broad fungal biomarker", rather than almost universal

- Study design: additional detail about the patient population at risk of IFD at this hospital. From the data, it appears that most cases are seen in Oncology. Were these only adult patients, or were paediatric patients included as well?

- Line 97- 98: elaborate on the negative samples chosen, since the EORTC/MSG guideline does not have a specific definition for what is considered "no evidence of IFD".

- Methods section: Serum BDG measurement- the fact that plasma is the samples of choice for the Wako assay should be mentioned to the reader in the methods, as well as the fact that a serum had not been validated on the assay before performance of this study. The only place this is mentioned is in the limitations section of the discussion.

- Line 263: hard to diagnose, not hard to diagnosis

- Additional limitations that require mentioning: the very small number of PJP cases.

- Table 1: Interquartile range should be added for patient age

- Line 341- patients "in" intensive care

- Table 2: It is not clear how the Wako median and IQR BDG value can be 0 (0- 0) for the group "No evidence of IFD", particularly if there were cases of false positives detecting value above 7 and 11pg/ml. This aspect requires re- analysis.

- Figure 1: requires axis titles on both x and y axes

Reviewer #3: De Carolis and colleagues present a comprehensive comparison of the Fungitell assay (FA) and the Glucan test (GT) representing the two CE certified beta-1,3-D-glucan assays (BDG). Four study cohorts are included in this study (patients suffering from invasive aspergillosis, invasive candidiasis, PJP, and a control group at risk for fungal infections but without evidence for fungal infections). The tests were performed in parallel and revealed astonishing high sensitivities and specificities. The authors demonstrate that sensitivities of GT for separate subgroups can be significantly further increased (without a major loss of specificity) by lowering the cut off.

This study enhances our knowledge about the performance of BDG diagnostics. The scientific community and all healthcare professionals benefit from this comparison of the two assays that are currently available in Europe.

However, there are some major concerns about this manuscript.

Major comments

1. The performance of BDG testing in this study is astonishing. The FA is commonly known to have the superior sensitivity but a lower specificity in comparison with GT (“FA weakness: specificity”). Contrarily, the GT is characterized by high specificity but inferior sensitivity compared to FA (“GT weakness: sensitivity”). However, in this study both assays excel not only in their strength but also in their putative weakness:

FA: The presented sensitivities (IA: 93 %, IC: 100 %, PJP: 100 %) are astonishing and (to my knowledge) the highest sensitivities published so far. Also the specificities (the “FA weakness”) are the best I did encounter until today (IA: 98 %, IC: 97 %; PJP: 97 %).

For comparison, I would like to refer to the results of different meta-analyses over the recent years, e.g., Onishi et al., 2011 (IFI sensitivity and specificity: 80 % and 82 %), Lu et al., 2011 (IFI sensitivity and specificity: 76 % and 85 %), Karageorgopoulos et al., 2011 (IFI sensitivity and specificity: 77 % and 85 %),…

GT: High specificity is a well known feature of the GT. However, the results of this study are still excelling: 100 % for IA, 100 % for IC, 100 % for PJP. Also the sensitivity is notable (the “GT weakness”), particularly in the setting of IC / candidemia: While other recent studies (Friedrich et al., 2018, Dichtl et al., 2018) found only sensitivities of 43 – 67 %, this study peaked with a never seen sensitivity of 91 %.

This absolutely raises the questions why De Carolis and colleagues experienced so much better results than virtually all other groups. The “discrepant” results of virtually all previous studies must be addressed in the manuscript and the reasons for this difference must be discussed.

2. There are several conclusions that are not speculative or euphemistic but wrong. The authors state that “GT performed as wells as the FA” (l38). This statement is based on results like sensitivities in the setting of IA of 92.5 % (FA) and 60 % (GT). 60 % is far from 92.5 %. L39: “Further studies are expected to definitely establish the equivalence of the two BDG assays.” This prognosis lacks a scientific basis. Concerning the comparability of the assays, the authors rely on the expression “perfect concordance regarding the results (positive / negative) obtained with the FA […] and the GT […]” (ll180-181). Then they demonstrate that this “perfect concordance” is characterized by the following results in the different subgroups: “IFD (91.8 % agreement)”, “IA (91.6 % agreement)”, “IC (95 % agreement)”, and “PJP (96 % agreement)”. The agreement is high, but perfect agreement means 100 %.

3. This study tends to favor the GT. Why was a ROC curve analysis only performed for the GT? Why were the results only reevaluated with an optimized cut off of the GT? What would happen to the results of FA testing when an optimized cut off was used?

This manuscript should be carefully revised in order to provide a neutral comparison of both tests that meets all scientific requirements.

Minor comments

The title suggests that PJP is not an invasive fungal infection. Why?

L24: The assays are not restricted to serum (plasma!).

L71: In which specificity did decreasing the cut off of the GT result?

L74: not

Ll139-141: This sentence (x samples of x cases, y samples of y cases, z samples of z cases) should be reworded.

Ll141-143: There is no EORTC/MSG definition for PJP.

Ll149-150: This sentence just reflects the results of the previous sentence.

Ll150-151: The first part of the sentence just reflects the results of the two previous sentences.

Ll209-211: There is no EORTC/MSG case definition for PJP. However, there is an EORTC/MSG case definition for proven invasive Candida infections: Cultivation of Candida from blood culture (= sterile body fluid) makes this infection a proven IC according to the criteria. Hence, the study of Friedrich et al. also relied on a large cohort of proven IC.

Ll211-215: This statement should be clarified.

Ll215-217: Particularly in combination with the previous statement concerning the study of Friedrich et al., this statement is very confusing. Why do the authors draw the conclusion that positive blood cultures might be a suboptimal gold standard?

L243: about the role of serum BDG (assay)

L247: could be the use of serum instead of plasma; ?

Reviewer #4: This manuscript studied the performance of the Wako Glucan test for the detection of beta-D-glucan (BDG) in patients with invasive fungal infections, and compared it to the Fungitel assay, which is currently the most widely used assay. The study was well designed and is appropriate for this scientific question. The sample size is sufficiently large with regards to IA and IC (40 and 78 cases respectively), but rather small for the PJP subgroup (only 17 cases). Overall, the manuscript is concise, well written and well structured.

We found only some minor issues:

1. English editing for spelling and grammar is required. For example, sentences like lines 255-257 are hard to read and understand, and use non-standard idioms.

2. How were patients selected? The authors mention that they selected patients with proven or probable IFD, which suggests a pre-existing register. The selection process can be biased depending on how it is performed (eg selection based on positive cultures invariably leads to higher biomarker levels, compared to selection based on PCR or other antigen tests). Please add this information to the M&M section.

3. The authors used the 2008 EORTC/MSG definition, likely because this study was conceived and performed prior to the release of the current 2019 revisions, which is entirely understandable. However, for clarity purposes, it seems best to at least acknowledge these new definitions. Furthermore, would it be possible to use the new classification for this manuscript instead? I believe this would require only a minor effort: for IC, no reclassification should be required as the criteria have been extended on top of the 2008 definitions. The classification used for PJP in this manuscript matches with EORTC/MSG proven or probable PJP (depending on positive IF or PCR). Only for IA, reclassification could be required depending on the result of serum or BALf GM, which can likely be done easily. Radiologic features and host features were extended, and therefore do not need revision for the 2019 definitions.

4. Please provide information on serum GM values in IA as a measure of fungal burden (see also comment 6)

5. The authors use a mixed control group of patients with different underlying diseases for all analyses. However, patients at risk for IC do not have the same risk factors as those at risk for IA (eg abdominal surgery is not a major risk factor for IA or PJP), which could skew the test parameters (eg use of surgical gauze during surgery is a known source of false positive BDG, which would not be present for patients at risk for IA or PJP). Was there a large difference in specificity in these control subgroups?

6. The sensitivity for BDG in IC and IA in this study is significantly higher than reported (see the meta-analyses by Onishi et al, 2011; Lu et al, 2011). What could be the reason for this according to the authors? Were patients diagnosed only in the later stages of infection (which leads to higher BDG levels)?

7. Table 1 or in text: please provide more info on the mycological features of the cases. How many were culture positive, how many GM, how many had positive IF or PCR in PJP? How many IC cases were blood culture positive, and how many histopathology, or both? This is relevant as BDG levels are generally higher in culture positive cases.

8. Table 2 appears to be a duplicate of what is already given in text. Please remove either one.

9. Line 141: please remove “proven” PJP when using your own definitions, as this could be confused with the 2019 defined proven PJP.

10. Section on performance of FA and GT (lines 145 and following): please provide p-values for all comparisons of diagnostic parameters between these two assays, using appropriate statistical methods. The authors compare sensitivity and specificity regularly throughout the manuscript and make claims about superiority of one over the other, without evaluating if this difference is statistically significant.

11. As the levels of BDG depend on the disease (higher in IC/PJP than in IA), it seems appropriate to define a cutoff for each. The authors even mention this themselves in the discussion, yet did not perform this relatively simple data exercise. Could they therefore verify of a different cutoff would be required for IA, compared to PJP/IC?

12. Lines 177-179: Is the difference between correlation coefficient statistically significantly different between the three diseases? They appear to be very close to each other.

13. Lines 209-2011: What do the authors mean? A positive blood culture can be considered as proven IC, so there should not be any difference there. Do the authors imply that, as they also included proven cases based on positive histopathology without positive blood culture in addition to only a positive blood culture, that this could be an explanation for a higher sensitivity in their study? This seems counter-intuitive. Comment 7 would also help in clarifying this issue.

14. Could the authors add a small explanation to the discussion or intro on why the numeric values are different between the FA and GT, while still testing for the same component (BDG) due to the use of different standards? This could otherwise be confusing for non-expert readers.

6. PLOS authors have the option to publish the peer review history of their article (what does this mean?). If published, this will include your full peer review and any attached files.

Reviewer #1: No

Reviewer #2: No

Reviewer #3: No

Reviewer #4: No

---

## [Author Response · Author response to Decision Letter 0]

28 May 2020

PONE-D-19-33241

A rebuttal letter that responds to each point raised by the academic editor and reviewer(s). This letter should be uploaded as separate file and labeled 'Response to Reviewers'.

A marked-up copy of your manuscript that highlights changes made to the original version. This file should be uploaded as separate file and labeled 'Revised Manuscript with Track Changes'.

An unmarked version of your revised paper without tracked changes. This file should be uploaded as separate file and labeled 'Manuscript'.

Please include your amended Competing Interests Statement within your cover letter.

5. Review Comments to the Author

Please use the space provided to explain your answers to the questions above. You may also include additional comments for the author, including concerns about dual publication, research ethics, or publication ethics. (Please upload your review as an attachment if it exceeds 20,000 characters).

Reviewer #1: This article addresses the diagnostic cutoffs for the available fungal diagnostic tests. It suggests new cutoffs that improve detection for those tests, which is in agreement with previous data. The change in cutoff is very small however. This article may be relevant for the field.

Answer: We thank the reviewer for his/her appreciation of our study. We agree that the change in the assay’s cutoff we proposed is small. However, we believe that this information may be relevant in the fungal diagnostics field.

Reviewer #2: Thank you for the opportunity to review this manuscript.

De Carolis and colleagues present a well-performed study demonstrating the performance of the Wako beta glucan assay in comparison to the Fungitell assay. Overall, the study design is robust and performance of the study is well explained.

English language editing can be sought to improve the readability of the manuscript.

Answer: We thank the reviewer for his/her appreciation of our study. As requested, the English language editing benefited from reviewing by an expert colleague.

Specific comments:

- Line 45: suggest reword the opening sentence to: "Invasive Aspergillosis, IC and PJP represent the most prevalent invasive fungal infections worldwide".

Answer: We reworded the opening sentence as requested.

- Line 46: reword "previously known as" to "also known as". PCP is still being used as a term for the disease.

Answer: We modified as suggested.

- Line 51: Suggest remove "e.g. respiratory"

Answer: The specification was removed.

- Line 51: non-specific, not unspecific

Answer: The term “unspecific” was changed to “nonspecific”.

- Line 54: Suggest reword- instead of unsuccessful, rather mention that culture methods have poor sensitivity or yield false negative results.

Answer: The sentence was modified as suggested.

- Line 56: reword as empiric treatment may be necessary. (the reference used here relates specifically to respiratory disease in patients with hematological disease). In many other cases of invasive fungal disease, it may not be appropriate to initiate empiric therapy.

Answer: We modified “is necessary” as “may be necessary”.

- Line 57: Reword as a "broad fungal biomarker", rather than almost universal.

Answer: We modified the term as suggested.

- Study design: additional detail about the patient population at risk of IFD at this hospital. From the data, it appears that most cases are seen in Oncology. Were these only adult patients, or were pediatric patients included as well?

Answer: Many cases were from oncology/hematology patients, and all were from adult patients. See the text and revised Table 1.

- Line 97-98: elaborate on the negative samples chosen, since the EORTC/MSG guideline does not have a specific definition for what is considered "no evidence of IFD".

Answer: We clarified that our cases did not meet the criteria for proven or probable disease and thus were considered as not having evidence of IFD.

- Methods section: Serum BDG measurement- the fact that plasma is the sample of choice for the Wako assay should be mentioned to the reader in the methods, as well as the fact that a serum had not been validated on the assay before performance of this study. The only place this is mentioned is in the limitations section of the discussion.

Answer: According to this comment, we added a sentence clarifying this issue in the M&M section.

- Line 263: hard to diagnose, not hard to diagnosis

Answer: The term “hard-to-diagnosis” was changed to “hard-to-diagnose”.

- Additional limitations that require mentioning: the very small number of PJP cases.

Answer: We added the additional limitation concerning the number of PJP cases.

- Table 1: Interquartile range should be added for patient age.

Answer: The interquartile range for patient age was added. See revised Table 1.

- Line 341- patients "in" intensive care.

Answer: “in” was corrected in the Table 1 footnote.

- Table 2: It is not clear how the Wako median and IQR BDG value can be 0 (0-0) for the group "No evidence of IFD", particularly if there were cases of false positives detecting value above 7 and 11 pg/ml. This aspect requires re-analysis.

Answer: We checked for correctness all the values previously presented in Table 2 (these values now appear in the text, because we deleted the original Table 2 as suggested by reviewer 4).

- Figure 1: requires axis titles on both x and y-axes.

Answer: Figure 1 was emended to show axis title on both x and y-axes. In addition, Figure 1 was modified, following the suggestion of reviewer 3, to show the results of the ROC analysis with an optimized cutoff also for the FA.

Reviewer #3: De Carolis and colleagues present a comprehensive comparison of the Fungitell assay (FA) and the Glucan test (GT) representing the two CE certified beta-1,3-D-glucan assays (BDG). Four study cohorts are included in this study (patients suffering from invasive aspergillosis, invasive candidiasis, PJP, and a control group at risk for fungal infections but without evidence for fungal infections). The tests were performed in parallel and revealed astonishing high sensitivities and specificities. The authors demonstrate that sensitivities of GT for separate subgroups can be significantly further increased (without a major loss of specificity) by lowering the cutoff.

This study enhances our knowledge about the performance of BDG diagnostics. The scientific community and all healthcare professionals benefit from this comparison of the two assays that are currently available in Europe.

However, there are some major concerns about this manuscript.

Major comments

1. The performance of BDG testing in this study is astonishing. The FA is commonly known to have the superior sensitivity but a lower specificity in comparison with GT (“FA weakness: specificity”). Contrarily, the GT is characterized by high specificity but inferior sensitivity compared to FA (“GT weakness: sensitivity”). However, in this study both assays excel not only in their strength but also in their putative weakness:

FA: The presented sensitivities (IA: 93 %, IC: 100 %, PJP: 100 %) are astonishing and (to my knowledge) the highest sensitivities published so far. Also the specificities (the “FA weakness”) are the best I did encounter until today (IA: 98 %, IC: 97 %; PJP: 97 %).

For comparison, I would like to refer to the results of different meta-analyses over the recent years, e.g., Onishi et al., 2011 (IFI sensitivity and specificity: 80 % and 82 %), Lu et al., 2011 (IFI sensitivity and specificity: 76 % and 85 %), Karageorgopoulos et al., 2011 (IFI sensitivity and specificity: 77 % and 85 %),..

Answer: We expanded the Discussion to contextualize our findings by referring to the results of several meta-analyses published over the recent years.

GT: High specificity is a well-known feature of the GT. However, the results of this study are still excelling 100 % for IA, 100 % for IC, 100 % for PJP. Also the sensitivity is notable (the “GT weakness”), particularly in the setting of IC / candidemia: While other recent studies (Friedrich et al., 2018, Dichtl et al., 2018) found only sensitivities of 43 – 67 %, this study peaked with a never seen sensitivity of 91 %.

This absolutely raises the questions why De Carolis and colleagues experienced so much better results than virtually all other groups. The “discrepant” results of virtually all previous studies must be addressed in the manuscript and the reasons for this difference must be discussed.

Answer: We expanded the Discussion to address the “discrepant” results of relevant previous studies in order to explain the reasons for this difference.

2. There are several conclusions that are not speculative or euphemistic but wrong. The authors state that “GT performed as well as the FA” (L38). This statement is based on results like sensitivities in the setting of IA of 92.5% (FA) and 60% (GT). 60% is far from 92.5 %. L39: “Further studies are expected to definitely establish the equivalence of the two BDG assays.” This prognosis lacks a scientific basis. Concerning the comparability of the assays, the authors rely on the expression “perfect concordance regarding the results (positive / negative) obtained with the FA […] and the GT […]” (L180-181). Then they demonstrate that this “perfect concordance” is characterized by the following results in the different subgroups: “IFD (91.8 % agreement)”, “IA (91.6 % agreement)”, “IC (95 % agreement)”, and “PJP (96 % agreement)”. The agreement is high, but perfect agreement means 100 %.

Answer: All the statements emphasizing the GT performance were mitigated as requested. Additionally, the term “perfect concordance” was changed to “excellent concordance” according to the definitions specified in M&M.

3. This study tends to favor the GT. Why was a ROC curve analysis only performed for the GT? Why were the results only reevaluated with an optimized cut off of the GT? What would happen to the results of FA testing when an optimized cut off was used?

Answer: To be absolutely impartial (see comment below), we performed a ROC curve also for the FA. See Results of the revised manuscript.

This manuscript should be carefully revised in order to provide a neutral comparison of both tests that meets all scientific requirements.

Answer: We revised the manuscript throughout to be somewhat neutral into comparing both FA and GT tests in order to satisfy all scientific requirements.

Minor comments

The title suggests that PJP is not an invasive fungal infection. Why?

Answer: The title was modified to avoid misunderstanding about PJP.

L24: The assays are not restricted to serum (plasma!).

Answer: The sentence was modified to also mention the plasma.

L71: In which specificity did decreasing the cutoff of the GT result?

Answer: This important detail was added.

L74: not

Answer: “not” was used instead of “no”.

L139-141: This sentence (x samples of x cases, y samples of y cases, z samples of z cases) should be reworded.

Answer: The sentence was amended.

L141-143: There is no EORTC/MSG definition for PJP.

Answer: According to this observation, we added the appropriate reference (Alanio et al. JAC 2016) for PJP.

L149-150: This sentence just reflects the results of the previous sentence.

Answer: The sentence was deleted.

L150-151: The first part of the sentence just reflects the results of the two previous sentences.

Answer: The first part of the sentence was deleted.

L209-211: There is no EORTC/MSG case definition for PJP. However, there is an EORTC/MSG case definition for proven invasive Candida infections: Cultivation of Candida from blood culture (= sterile body fluid) makes this infection a proven IC according to the criteria. Hence, the study of Friedrich et al. also relied on a large cohort of proven IC.

Answer: The sentence was modified to clarify that 2008 EORTC/MSG definitions do not include PJP as well as to precise that blood culture is really a proven IC criterion.

Ll211-215: This statement should be clarified.

Answer: The statement was modified for clarity.

Ll215-217: Particularly in combination with the previous statement concerning the study of Friedrich et al., this statement is very confusing. Why do the authors draw the conclusion that positive blood cultures might be a suboptimal gold standard?

Answer: The statement was modified to improve the clarity.

L243: about the role of serum BDG (assay)

Answer: The term assay was deleted.

L247: could be the use of serum instead of plasma?

Answer: The sentence was amended.

Reviewer #4: This manuscript studied the performance of the Wako Glucan test for the detection of beta-D-glucan (BDG) in patients with invasive fungal infections, and compared it to the Fungitell assay, which is currently the most widely used assay. The study was well designed and is appropriate for this scientific question. The sample size is sufficiently large with regards to IA and IC (40 and 78 cases respectively), but rather small for the PJP subgroup (only 17 cases). Overall, the manuscript is concise, well written and well structured.

We found only some minor issues:

1. English editing for spelling and grammar is required. For example, sentences like lines 255-257 are hard to read and understand, and use non-standard idioms.

Answer: As requested, the English language editing benefited from reviewing by an expert colleague.

2. How were patients selected? The authors mention that they selected patients with proven or probable IFD, which suggests a pre-existing register. The selection process can be biased depending on how it is performed (e.g. selection based on positive cultures invariably leads to higher biomarker levels, compared to selection based on PCR or other antigen tests). Please add this information to the M&M section.

Answer: Details about the selection of patients with proven or probable IFD were provided in the M&M section.

3. The authors used the 2008 EORTC/MSG definition, likely because this study was conceived and performed prior to the release of the current 2019 revisions, which is entirely understandable. However, for clarity purposes, it seems best to at least acknowledge these new definitions. Furthermore, would it be possible to use the new classification for this manuscript instead? I believe this would require only a minor effort: for IC, no reclassification should be required as the criteria have been extended on top of the 2008 definitions. The classification used for PJP in this manuscript matches with EORTC/MSG proven or probable PJP (depending on positive IF or PCR). Only for IA, reclassification could be required depending on the result of serum or BALF GM, which can likely be done easily. Radiologic features and host features were extended, and therefore do not need revision for the 2019 definitions.

Answer: We added a sentence stating the status of IFD cases according to the 2019 EORTC/MSG definition criteria. See the M&M section of the revised manuscript.

4. Please provide information on serum GM values in IA as a measure of fungal burden (see also comment 6)

Answer: Information about the serum GM values in IA was provided. See Discussion and the new Table 2.

5. The authors use a mixed control group of patients with different underlying diseases for all analyses. However, patients at risk for IC do not have the same risk factors as those at risk for IA (e.g. abdominal surgery is not a major risk factor for IA or PJP), which could skew the test parameters (e.g. use of surgical gauze during surgery is a known source of false positive BDG, which would not be present for patients at risk for IA or PJP). Was there a large difference in specificity in these control subgroups?

Answer: As suggested, we performed a rough analysis using control subgroups to assess potential differences in specificity among the patients at risk for a different IFD.

6. The sensitivity for BDG in IC and IA in this study is significantly higher than reported (see the meta-analyses by Onishi et al, 2011; Lu et al, 2011). What could be the reason for this according to the authors? Were patients diagnosed only in the later stages of infection (which leads to higher BDG levels)?

Answer: A comment about the possible reasons for the high sensitivity seen in our study was added.

7. Table 1 or in text: please provide more info on the mycological features of the cases. How many were culture positive, how many GM, how many had positive IF or PCR in PJP? How many IC cases were blood culture positive, and how many histopathology, or both? This is relevant as BDG levels are generally higher in culture positive cases.

Answer: We added a new Table 2 (in substitution of the original Table 2, which was deleted; see comment below) to include all the information about mycological features of the cases.

8. Table 2 appears to be a duplicate of what is already given in text. Please remove either one.

Answer: Table 2 was deleted.

9. Line 141: please remove “proven” PJP when using your own definitions, as this could be confused with the 2019 defined proven PJP.

Answer: “proven” was removed when referring to as PJP.

10. Section on performance of FA and GT (lines 145 and following): please provide p-values for all comparisons of diagnostic parameters between these two assays, using appropriate statistical methods. The authors compare sensitivity and specificity regularly throughout the manuscript and make claims about superiority of one over the other, without evaluating if this difference is statistically significant.

Answer: Our study aimed to evaluate two BDG assays in parallel. According to the main comments of reviewer 3, we omitted claims about the superiority of one over the other. Therefore, a statistical comparison of these differences was waived.

11. As the levels of BDG depend on the disease (higher in IC/PJP than in IA), it seems appropriate to define a cutoff for each. The authors even mention this themselves in the discussion, yet did not perform this relatively simple data exercise. Could they therefore verify of a different cutoff would be required for IA, compared to PJP/IC?

Answer: As suggested, we performed additional analyses to verify the potentiality of a different cutoff for IA compared to PJP/IC. See Results of the revised manuscript.

12. Lines 177-179: Is the difference between correlation coefficient statistically significantly different between the three diseases? They appear to be very close to each other.

Answer: We checked that the differences were statistically significant.

13. Lines 209-211: What do the authors mean? A positive blood culture can be considered as proven IC, so there should not be any difference there. Do the authors imply that, as they also included proven cases based on positive histopathology without positive blood culture in addition to only a positive blood culture, that this could be an explanation for a higher sensitivity in their study? This seems counter-intuitive. Comment 7 would also help in clarifying this issue.

Answer: The sentence was modified to improve clarity and to add relevant information, also in accordance with what suggested in comment 7.

14. Could the authors add a small explanation to the discussion or intro on why the numeric values are different between the FA and GT, while still testing for the same component (BDG) due to the use of different standards? This could otherwise be confusing for non-expert readers.

Answer: A possible explanation in the Introduction section was added.

---

## [Decision Letter · Decision Letter 1]

15 Jun 2020

PONE-D-19-33241R1

Comparative performance evaluation of Wako β-glucan test and Fungitell assay for the diagnosis of invasive fungal diseases

PLOS ONE

Dear Dr. Sanguinetti,

Thank you for submitting your manuscript to PLOS ONE. After careful consideration, we feel that it has merit but does not fully meet PLOS ONE’s publication criteria as it currently stands. Therefore, we invite you to submit a revised version of the manuscript that addresses the points raised during the review process.

Please address the final reviewer concerns.

We look forward to receiving your revised manuscript.

Kind regards,

Jeffrey Chalmers, Ph.D.

Academic Editor

PLOS ONE

Reviewers' comments:

Reviewer's Responses to Questions

**Comments to the Author**

1. If the authors have adequately addressed your comments raised in a previous round of review and you feel that this manuscript is now acceptable for publication, you may indicate that here to bypass the “Comments to the Author” section, enter your conflict of interest statement in the “Confidential to Editor” section, and submit your "Accept" recommendation.

Reviewer #2: All comments have been addressed

Reviewer #4: (No Response)

2. Is the manuscript technically sound, and do the data support the conclusions?

Reviewer #2: Yes

Reviewer #4: Yes

3. Has the statistical analysis been performed appropriately and rigorously? 

Reviewer #2: Yes

Reviewer #4: No

4. Have the authors made all data underlying the findings in their manuscript fully available?

Reviewer #2: Yes

Reviewer #4: Yes

5. Is the manuscript presented in an intelligible fashion and written in standard English?

Reviewer #2: Yes

Reviewer #4: Yes

6. Review Comments to the Author

Reviewer #2: (No Response)

Reviewer #4: Most of my comments have been addressed. However, I disagree with the reply to comment 10 regarding statistical analysis. Overall, Reviewer #3's comments are largely the same as mine. We both state that you cannot make any proclamation about superiority without doing due analysis. However, this does not mean this is not possible, or indeed, that this can be waived!

The fact that this is a head-to-head analysis means a pairwise comparison is indicated, such as McNemar's test. Using this test, you can confidently state for example, that the sensitivity of FA > 80 is significantly greater than the sensitivity of GT > 7 (92.5% vs 80%, p=0.025). Most statistical packages have tools specifically for comparing 2 diagnostic tests.

7. PLOS authors have the option to publish the peer review history of their article (what does this mean?). If published, this will include your full peer review and any attached files.

Reviewer #2: No

Reviewer #4: No

---

## [Author Response · Author response to Decision Letter 1]

17 Jun 2020

PONE-D-19-33241R1

Reviewer #4: Most of my comments have been addressed. However, I disagree with the reply to comment 10 regarding statistical analysis. Overall, Reviewer #3's comments are largely the same as mine. We both state that you cannot make any proclamation about superiority without doing due analysis. However, this does not mean this is not possible, or indeed, that this can be waived!

The fact that this is a head-to-head analysis means a pairwise comparison is indicated, such as McNemar's test. Using this test, you can confidently state for example, that the sensitivity of FA > 80 is significantly greater than the sensitivity of GT > 7 (92.5% vs 80%, p=0.025). Most statistical packages have tools specifically for comparing two diagnostic tests.

Answer: We are delighted in noticing that we were able to satisfy most of the reviewer’s comments. Regarding the unsatisfied remaining comment raised by the reviewer (also in line with the reviewer #3), we treasured the suggestion of the reviewer. Thus, we used the McNemar's test to assess statistically the differences in performance parameters between FA and GT. Accordingly, we added five sentences, one of which in the Materials & Methods section (Statistical Analysis paragraph) and other four in the Results section (Performances of the FA and the GT). See the revised version of the manuscript (PONE-D-19-33241R1 with Track Changes).

---

## [Decision Letter · Decision Letter 2]

30 Jun 2020

Comparative performance evaluation of Wako β-glucan test and Fungitell assay for the diagnosis of invasive fungal diseases

PONE-D-19-33241R2

Dear Dr. Sanguinetti,

We’re pleased to inform you that your manuscript has been judged scientifically suitable for publication and will be formally accepted for publication once it meets all outstanding technical requirements.

Kind regards,

Jeffrey Chalmers, Ph.D.

Academic Editor

PLOS ONE

Additional Editor Comments (optional):

Reviewers' comments:

Reviewer's Responses to Questions

**Comments to the Author**

1. If the authors have adequately addressed your comments raised in a previous round of review and you feel that this manuscript is now acceptable for publication, you may indicate that here to bypass the “Comments to the Author” section, enter your conflict of interest statement in the “Confidential to Editor” section, and submit your "Accept" recommendation.

Reviewer #4: All comments have been addressed

2. Is the manuscript technically sound, and do the data support the conclusions?

Reviewer #4: (No Response)

3. Has the statistical analysis been performed appropriately and rigorously? 

Reviewer #4: (No Response)

4. Have the authors made all data underlying the findings in their manuscript fully available?

Reviewer #4: (No Response)

5. Is the manuscript presented in an intelligible fashion and written in standard English?

Reviewer #4: (No Response)

6. Review Comments to the Author

Reviewer #4: (No Response)

7. PLOS authors have the option to publish the peer review history of their article (what does this mean?). If published, this will include your full peer review and any attached files.

Reviewer #4: No

---

## [Editor Report · Acceptance letter]

17 Jul 2020

PONE-D-19-33241R2 

Comparative performance evaluation of Wako β-glucan test and Fungitell assay for the diagnosis of invasive fungal diseases 

Dear Dr. Sanguinetti:

I'm pleased to inform you that your manuscript has been deemed suitable for publication in PLOS ONE. Congratulations! Your manuscript is now with our production department. 

Kind regards, 

on behalf of

Dr. Jeffrey Chalmers 

Academic Editor

PLOS ONE